# The Heat Is On: Modeling the Persistence of ESBL-Producing *E. coli* in Blue Mussels under Meal Preparation

**DOI:** 10.3390/foods12010014

**Published:** 2022-12-20

**Authors:** Kyrre Kausrud, Taran Skjerdal, Gro S. Johannessen, Hanna K. Ilag, Madelaine Norström

**Affiliations:** 1Norwegian Veterinary Institute, 1431 Ås, Norway; 2Department of Microbiology, Oslo University Hospital, 0424 Oslo, Norway

**Keywords:** heat treatment, *E. coli*, mussels, ESBL, AMR, exposure models

## Abstract

Pathways for exposure and dissemination of antimicrobial-resistant (AMR) bacteria are major public health issues. Filter-feeding shellfish concentrate bacteria from the environment and thus can also harbor extended-spectrum β-lactamase—producing *Escherichia coli* (ESBL *E. coli*) as an example of a resistant pathogen of concern. Is the short steaming procedure that blue mussels (*Mytilus edulis*) undergo before consumption enough for food safety in regard to such resistant pathogens? In this study, we performed experiments to assess the survival of ESBL *E. coli* in blue mussel. Consequently, a predictive model for the dose of ESBL *E. coli* that consumers would be exposed to, after preparing blue mussels or similar through the common practice of brief steaming until opening of the shells, was performed. The output of the model is the expected number of colony forming units per gram (cfu/g) of ESBL *E. coli* in a meal as a function of the duration and the temperature of steaming and the initial contamination. In these experiments, the heat tolerance of the ESBL-producing *E. coli* strain was indistinguishable from that of non-ESBL *E. coli*, and the heat treatments often practiced are likely to be insufficient to avoid exposure to viable ESBL *E. coli*. Steaming time (>3.5–4.0 min) is a better indicator than shell openness to avoid exposure to these ESBL or indicator *E. coli* strains.

## 1. Introduction

Shellfish such as blue mussels are consumed after a short heat treatment. As shellfish filter seawater, they accumulate microbes and the effectivity of the heat treatment for decontamination is therefore critical for food safety.

Antimicrobial-resistant (AMR) bacteria in seafood represent a potential risk for human beings by two main mechanisms, either clonal transfer of resistant bacteria or by horizontal gene transfer (HGT) of mobile genetic elements (MGEs) to previously susceptible pathogens. The emergence of successful multi-drug-resistant (MDR) variants of *Escherichia coli* and *Klebsiella pneumoniae*, belonging to certain clonal lineages, has contributed to the rapid global spread of extended-spectrum beta lactamase-producing Gram negative bacteria (ESBLs) and carbapenemases [1]. These clones are considered “global high risk clones” and have an excellent ability to colonize human hosts, disseminate and cause infections, with *E. coli* Sequence type (ST) 131 and *K. pneumoniae* ST258 as pertinent examples [2]. To be better equipped for the emerging AMR challenge, a thorough investigation of transmission routes and reservoirs is needed. WHO underlines the knowledge gap of the food chain in transmission of AMR bacteria, and AMR bacteria from seafood have been identified by EFSA as an issue for monitoring [3]. ESBL-producing *E. coli* is one of several emerging AMR microbes that have been detected in blue mussels [4,5]. The origin of such resistances may be both from human or animal sources contaminating seawater [4]. The filtration rate of water in blue mussels is temperature dependent, but at 15 °C it may exceed 120 L of water per day [6]. They therefore constitute potential hot spots for accumulating pathogenic and antimicrobial-resistant bacteria from the marine environment. If ESBL organisms accumulated from the environment survive the light–heat treatment that is traditionally preferred for mussels before consumption, AMR genetic elements can be transferred to the human microbial community. Few prevalence studies have estimated the occurrence of AMR in blue mussels; however, the occurrence of AMR in shellfish will most probably reflect the occurrence in the environment where they have been grown [4,5,7,8].

The potential for blue mussels to be a significant source of ESBL- producing *E. coli* to human beings is unknown. A study performed under the Norwegian monitoring program for antimicrobial resistance in the veterinary and food production sectors (NORM-VET) in 2016 reported that 4.2% of *E. coli* isolates obtained from bivalve molluscs (*n* = 261) in Norway were resistant to at least one antibiotic, while the prevalence of resistance to three or more antibiotics was 1.4% [9]. By using a selective screening methodology, 3.3% of the 391 samples showed resistance to third-generation cephalosporins and ten of these carried the globally common plasmid-encoded ESBL resistance gene blaCTX-M-15 [9].

Shellfish such as oysters and mussels are often consumed raw or undercooked for culinary reasons, which may pose a risk for the consumer [10]. In addition, consumption of wild-harvested mussels, i.e., uncontrolled mussels, occurs in many coastal areas, particularly during vacation times. In these cases, the heat treatment is the only hurdle for ESBL exposure as the local contamination levels may be unknown or disregarded. Commercially produced blue mussels have a food safety regulation limit of maximum 10 *E. coli*/g by the end of manufacturing process for direct human consumption (Commission regulation (EC) No 2073/2005). Class A shellfish by harvest should not contain more than 230 cfu/g of *E. coli* by harvest. A recommended practice has been to move shellfish with higher *E. coli* concentrations to cleaner water until the concentration falls below this level or they will undergo an industrial heat treatment before being marketed [11]. Whether adhering to the required limit is enough to avoid ESBL *E. coli* exposure may depend on their initial concentration and how well they survive heat treatment.

It is therefore a need for knowledge about the trade-off between safety and preferred sensory quality, i.e., the potential for survival of both *E. coli* and ESBL-producing *E. coli* and minimum heating conditions for elimination of these microbes in shellfish. There exists little knowledge about the persistence of viable ESBL-producing *E. coli* in different food matrixes where only light–heat treatments are performed before consumption, but both the maximum obtained temperature within the mussels as well as the duration of certain temperatures will likely have an impact on bacterial survival rates

The aim of this study was therefore to assess the survival of ESBL-producing *E. coli* in blue mussels following different heat treatment regimes, and to develop a corresponding exposure model and tool for risk assessment and guideline development. To achieve this, we conducted experiments inoculating live shellfish with *E. coli* as an indicator for ESBL-producing *E. coli* to avoid unacceptable contamination risks and used ESBL-producing *E. coli* in controlled heat treatment experiments comparing it to *E. coli* to verify their role as an indicator and accumulate additional data on heat inactivation.

The model for the mussel production chain developed in this work incorporates two sets of experiments. The first set of experiments involves inoculating live mussels with non-ESBL-producing *E. coli* by allowing them to naturally filter contaminated water in an aquarium. This experiment could, however, not utilize ESBL strains due to biohazard procedures, and thus a second set of experiments addressed this by homogenizing a mix of mussel flesh and either non-ESBL or ESBL-producing *E. coli* in a series of heat-resistant plastic bags.

## 2. Materials and Methods

Blue mussels for the study were purchased at a local supermarket, where they had been stored on ice. The mussels were transported to the lab and within 45 min placed at 5 °C or in the aquarium for acclimatization.

Our approach consisted of two main steps:

Inoculating live blue mussels with *E. coli* in an aquarium and then steaming them for different lengths of time in a kettle, thus closely simulating normal consumer pro-cedure and materials. (See Section 2.1, Section 2.2, Section 2.3 and Section 2.5).

Inoculating raw de-shelled blue mussel flesh with non-ESBL-producing (indicator) *E. coli* and an ESBL-producing *E. coli*, respectively, in sealed plastic bags and subjecting them to heat treatments at various durations and controlled temperatures in water baths (See Section 2.1, Section 2.2, Section 2.4 and Section 2.5).

All experiments and analyses were conducted at the Norwegian Veterinary Insti-tute facilities in Oslo.

### 2.1. Preparation of Inoculum

For the inoculation studies described below, indicator *E. coli* (three isolates, 2016-22-55-1-1-1-1, 2016-01-4162-1-1-1-1 and 2016-01-4220-1-1-1-1) and ESBL-producing *E. coli* (ESBL) (two isolates, 2016-01-4162-1-3-1-1 and 2016-01-4220-1-3-1-1) were used. The five isolates were all isolated from blue mussels analyzed previously [9] and kindly provided by NORM-VET.

One isolate of indicator *E. coli* (2016-22-55-1-1-1-1) originating from a blue mussel purchased from a retail store was used for inoculation of the water in the aquarium experiment. Briefly, the inoculum was prepared from frozen glycerol stocks where a loopful (1 µL) was plated directly from the stock onto a blood agar plate (bovine blood) that was incubated at 37 ± 1 °C overnight. A single colony of *E. coli* from the blood agar was added to 100 mL Buffered Peptone Water (BPW-ISO, OXOID) and incubated over night at 37 ± 1 °C. The overnight broth culture was equally distributed in four 50 mL sterile tubes and washed twice by centrifugation for 10 min at 3800 g (Beckman GS-15R Centrifuge), removal of the supernatant and resuspension of the bacterial pellet in 10 mL 0.9% saline water. After the second wash, the pellets were resuspended in 10 mL 0.9% saline separately prior to adding to the aquarium.

For the experimental inoculation of blue mussel flesh, samples were spiked with two isolates of indicator *E. coli* (2016-01-4162-1-1-1-1 and 2016-01-4220-1-1-1-1) and two strains of ESBL-producing *E. coli* (2016-01-4162-1-3-1-1 and 2016-01-4220-1-3-1-1), respectively. Both ESBL-producing *E. coli* harbored blaCTX-M-15, where strain 2016-01-4162-1-3-1-1 harbored blaCTX-M-15 alone, while strain 2016-01-4220-1-3-1-1 also had blaCMY-2. These isolates originated from two samples in which both an indicator *E. coli* as well as an ESBL isolate had been detected through selective screening within the NORM-VET program. The inocula were prepared from frozen glycerol stocks by plating a loopful of stock on blood agar plates followed by incubation at 37 ± 1 °C overnight. A single colony from each isolate was transferred to separate tubes of 10 mL BPW-ISO and incubated as described above. After incubation, the two isolates of indicator *E. coli* and the two strains of ESBLs *E. coli* were mixed to an *E. coli* mix and an ESBL -*E. coli* mix, respectively, and used for direct inoculation of samples of blue mussel flesh.

### 2.2. Aquarium Experiment

Artificial seawater (3%) was made by adding 2100 g Red Sea Salt (© 2020 Red Sea) to approximately 58 L ice and cold 35 L tap water to an aquarium (equipped with an electric pump, clean but not sterile). The aquarium was situated in a room with a constant temperature of 16 °C. Immediately after the preparation of the artificial seawater, a total of 90–100 blue mussels were transferred to the aquarium for acclimation for 24 h before 40 mL inoculum of ~2 × 10^8^ pr mL *E. coli* (2016-22-55-1-1-1-1) was added. The aquarium experiment was carried out three times.

Water samples were taken after the *E. coli* overnight broth culture was added and before the blue mussels were harvested 17 h after inoculation. After being removed from the aquarium the mussels were brought to the laboratory on ice. Each was marked with a waterproof marker and kept at 5 °C until steaming. Uninoculated blue mussels with temperature loggers (Signatrol SL53T Temperature logger 0/125 °C) were included in the pot during steaming in order to obtain information on the temperature inside the mussel shells flesh during the cooking period. 

There was no growth of *E. coli* from the negative controls. A total of 10 inoculated blue mussels were transferred to a pot with approximately 70–80 not-inoculated blue mussels and steamed for pre-determined periods of time (30 s intervals from 60 to 210 s). After steaming, the flesh from the inoculated blue mussels were allowed to cool in room temperature before being distributed in Stomacher bags, one bag per mussel, for quantitative analyses of *E. coli* as described in Section 2.4 below. A total of 10 uninoculated blue mussels and 10 inoculated blue mussels prior to steaming were also distributed in Stomacher bags, one bag per mussel, for quantitative analysis for *E. coli* as negative controls and to check the level of *E. coli* present in the inoculated mussels, respectively.

### 2.3. Heat Treatment with Inoculated Blue Mussel Flesh

For the experiments with the heat treatment of the inoculated flesh from blue mussels, the flesh was obtained from blue mussels purchased from a local supermarket (see above). Aliquots of 10 g were distributed in separate 400 mL Stomacher bags (Grade Blender Bags, Standard 400) and inoculated separately with 100 µL of ESBL mix (2016-01-4162-1-3-1-1 and 2016-01-4220-1-3-1-1) or *E. coli* (2016-01-4162-1-1-1-1 and 2016-01-4220-1-1-1-1) mix (prior to heat sealing of the bags (for concentration of bacteria in the inocula). Before sealing, as much air as possible was squeezed out of the bags, and the mussel flesh was not flattened beyond this before heat treatment, retaining the effect of slow heat transfer within the mussel during cooking. The inoculated bags were then stored at 5 °C for at least 30 min prior to heat treatment. The heat treatment was carried out by completely submerging the parts of the bags with blue mussel flesh in a water bath (Nüve BM 30) at fixed temperatures (55 °C, 65 °C and 75 °C) for pre-determined periods of time (20–270 s). The bags were allowed to cool at room temperature analogous to the whole mussels prior to quantitative analyses for *E. coli* and ESBL, respectively, as described in Section 2.4.

Temperature loggers were included in similar samples with uninoculated blue mussel flesh.

Inoculated samples were analyzed for *E. coli* and ESBL *E. coli* prior to heat treatment to estimate the initial concentration in the mussel flesh. Uninoculated control samples without heat treatment were analyzed for *E. coli* and ESBL.

### 2.4. Microbiological Analyses

#### 2.4.1. Quantitative Analyses of Indicator *E. coli* and ESBL-Producing *E. coli*

The blue mussel flesh was weighed and diluted 1:10 by adding BPW-ISO to the Stomacher bag, prior to stomaching or shaking for 30 s to two minutes to homogenize. The samples were further serially diluted in BPW-ISO (aquarium experiment) or Peptone Saline (1 g peptone, 8.5 g NaCl/L) (heat treatment with inoculated blue mussel flesh) and 100 µL of the appropriate dilutions was plated out with a L-rod on TBX (Oxoid) or MacConkey (Becton Dickinson) supplemented with 1 mg/L cefotaxime (Sigma) (MaC-CO) for quantification of *E. coli* or ESBL-producing *E. coli*, respectively. In order to obtain a detection limit of 10 cfu/g, one mL of the initial homogenate was distributed equally on the surface of three plates. The TBX and MaC-CO plates were incubated at 37 ± 1 °C and 41.5 ± 1 °C, respectively. Typical colonies on the different agars were counted and a selection of colonies was further confirmed (Section 2.4.3).

#### 2.4.2. Detection of *E. coli* and ESBL-Producing *E. coli* in Enrichment of Samples

After serial dilutions and plating had been made for quantitative analysis, the rest of the initial homogenate (10 g sample and 90 mL BPW-ISO) from the heat treatment experiment and the aquarium experiment (flesh from one blue mussel diluted 1:10 with BPW-ISO) were incubated at 37 ± 1 °C overnight, followed by plating of a loopful (10 µL) of enrichment on TBX or MaC-CO, depending on which organism analyzed for. All samples were enriched, but plating was only performed if the result from the quantitative analysis was below the detection limit (i.e., <10 cfu/g). The plates were inspected for typical colonies and a selection of colonies was pure-cultured on blood agar and further confirmed (Section 2.4.3).

#### 2.4.3. Confirmation by MALDI-TOF and PCR

A selection of colonies from both the blue mussel inoculation experiment and the heat treatment experiment were confirmed as *E. coli* by MALDI Biotyper MS (MALDI-TOF MS, Bruker Daltronics GmbH). Presumptive ESBL-producing *E. coli* were confirmed by using PCR specific for the genes harbored by the strains [11,12,13].

### 2.5. Statistical Analysis and Modeling

Data analysis was conducted using the R 3.5 software [14] with the mgcv package for generalized additive models (GAMs) with smoothness estimators, and application of results was performed using R Shiny [15,16].

We first used a binomial model of steam time vs. probability *P* of mussels opening so that the estimate proportion *EP* of mussels opened for a given time point (Figure 1):(1)$$\mathrm{EP}(\mathrm{Open}|\mathrm{Time})=\frac{1}{1+e^{f\left(T\right)}}$$

Data were treated to account for “worst case scenarios”, so when *E. coli* were not detected by enumeration (i.e quantitative analysis with detection level 10 cfu/g), but only after enrichment, *E. coli* numbers were set to 9 cfu/g.

When modeling how many bacteria that would remain viable after heat treatment, repeated measures on the same mussel were not feasible. Hence, estimates of contamination level had to be made independently on different individual samples. As a measure Pe of proportional survival of bacteria, this means taking the cfu/g for a given sample at time t minutes of the heat treatment and dividing it by the average cfu/g found in inoculated samples at t = 0, i.e., before any heat above room temperature was applied, rounding any number
(2)$$P_{CFU,t}=\min\left\{\frac{cfu_{t}}{cfu_{0}},1\right\}$$
as we are assuming no significant further bacterial growth happening in the few minutes between initial samples being taken and heat treatment being applied, and thus sample variance being the cause for any number over 1.

The proportion of the remaining viable cfus were estimated by the GAM regression models with the non-parametric penalized thin spline model with quasi-binomial error distribution to allow for overdispersion and allow the estimation of survival curves to be data-driven and flexible rather than being bound to specific a priori survival functions such as the Weibull [17] or exponential. Logistic regressions on the proportion as given in the results and discussion all follow the general format of
(3)$$Y=\ln\left(\frac{1}{1-P_{CFU}}\right)$$
Y= b + B(X) + f(X,k) + ε(4)
where b is a constant (intercept), B a vector of constants and X a matrix of explanatory variables. f(X) denotes a set of zero or more penalized regression splines [18] with smoothing parameters selected by the GCV criterion limited upwards to a maximum number of degrees of freedom k, and the conditional distribution of the response a quasi-binomial distribution
(5)$$P\left(Y=k\right)=\left(\begin{array}{c} n\\ k \end{array}\right)p{\left(p+k\phi \right)}^{k-1}{\left(1-p-k\phi \right)}^{n-k\;}\;$$
which is similar to the binomial distribution except for the parameter ϕ, which captures excess variance. Some models incorporate only linear predictors B(X), others non-linear effects (f(X,k)), and this is made clear in the text for each relevant model.

When modeling the number of cfu/g directly, not as a proportion remaining of the initial concentration after inoculation, the same model framework was used. Except for the response variable
(6)$$Y=\ln\left(cfu\right)$$

And when the conditional distribution of the response is a quasi-Poisson distribution [19], i.e., where if
(7)$$E\left(Y\right)=\mu$$
(8)$$Var\left(Y\right)=\theta \mu$$
(9)$$P\left(Y=k\right)=\frac{\lambda^{k}e^{-\lambda }}{k!}$$
making it a Poisson distribution with an overdispersion parameter θ regulates the variance/mean relationship.

When estimating the exposure, we assumed that the observed concentrations of bacteria were representative of an underlying probability density. We then estimated smoothed empirical probability densities on the observed concentrations and simulated expected exposures by drawing 20 hypothetical shells as a typical meal from these distributions, taking the average bacterial concentration (cfu/g) and multiplying by 250, as 250 g is assumed to be a typical portion of blue mussels.

For temperature logger data, a set of algorithms was developed to identify and homogenize logger time series, but an element of manual delineating was most efficient as some treatment times had not left peaks identifiable beyond noise and temperature fluctuations between refrigerators, water and air.

## 3. Results

### 3.1. Experiments

In traditional preparation, where a culinary value is placed on minimizing heat exposure, looking for the mussel shells opening under steaming is a common indicator for when they can be taken off the heat. The opening rate is well described by a logistic model of opening as a function of the steaming time T, which explains about 70% of the variance in openness status for mussels (Methods Section 2.2 and Equation (1), Figure 1). The opening rate as a functioning of heat (i.e., steam) exposure was consistent between experiments, and showed no statistically significant differences between steaming batches.

Black squares are averages. *y*-axis values represent the fraction of mussels open (i.e., 0 = all closed; 1 = all open). The blue rug lines indicate datapoints along the time axis. Model fit is shown in red, with a 2SE (SE = Standard error) confidence interval for model fit shaded gray.

Whether or not a mussel is fully open is highly correlated with the proportion *P_CFU_* (see Section 2.5 of *E. coli* being cultivable from that mussel, but alone it explains only about 40% of the variance in the proportion of bacteria being viable after steaming. Including the starting concentration (mean cfu/g in samples from the same batch taken before steaming) as an explanatory variable was not significant, suggesting that at these concentrations the survival rate of bacteria was independent of concentration. See Figure 2. We see a strong reduction in average bacterial concentrations as shells open, but also that even some opened shells retain fairly high bacterial concentrations.

Simulating meal exposures from eating 20 mussels (assumed to be a typical meal) suggests a 10% risk of ingesting a dose over 10% of the original (pre-steaming) concentration of bacteria. If partially opened mussels are included in the meal, the risk increases significantly, as 10% of meals will contain bacteria corresponding to 20% or more of the original concentration in the meal as a whole. See Figure 3.

#### 3.1.1. Steaming Time Effects on Bacterial Concentration

Modeling the cfu/g directly in an overdispersed Poisson regression model (see Section 2.5) using the steaming time and the mean initial bacterial concentration in the un-steamed mussels (cfug0) as the variables explains about 60% of the variance. The remaining variation is likely to be due to the cooling period after the steaming and until the analysis of the sample, and to the random variation in the sampling and culturing. The open status loses all significant explanatory power when the steaming time is allowed to enter as a non-linear effect (see Section 2.5). After the mussels had been steamed for >3.5 min, *E. coli* was not detected in any of the samples. We thus obtained a range of steam times predicted to bring exposure down to regulation levels depending on meal size and contamination level estimated from this. See Figure 4.

These steam times map closely to the maximum temperature found by temperature loggers to be attained within steamed mussels and exchanging the max temperature for the time had the same explanatory power. The effect of the steaming time is thus suggested to be largely mediated through the maximum temperature and the time over which it is applied. However, using maximum temperature as an explanatory variable does suggest that the effect of temperature is strongly non-linear and that the *E. coli* strains used in the present study start to be inactivated at temperatures exceeding 55°C. See Figure 4.

#### 3.1.2. Heat Treatment

For comparisons of indicator strains of *E. coli* and ESBL producing *E. coli*, experiments spanning the temperatures relevant for steamed shells were chosen. Mussel flesh was therefore taken out of the shell prior to heating, and the mussel flesh was inoculated with the bacteria.

Blue mussel flesh was then inoculated directly and treated at constant temperatures for pre-determined periods. 

At 55 °C, both ESBL-producing *E. coli* and indicator *E. coli* remained in high concentrations even after 270 seconds of heat treatment. Surprisingly, the germination rate seemed possibly even higher after warming. 

At 65°C, there was more variation, for both indicator *E. coli* and ESBL-producing *E. coli* being detected in some of the samples that were treated for 90 to 240 s, but after 270 s at 65 °C, neither indicator *E. coli* nor ESBL-producing *E. coli* could be detected. 

At 75 °C, the mussel flesh samples were generally negative after 40–60 s treatment, but some *E. coli* and ESBL-producing *E. coli* could be detected in some of the samples up to 100 s treatment. This could be explained by the uneven distribution of the inoculum in the samples. After 110 s at 75 °C, neither indicator *E. coli* nor ESBL-producing *E. coli* could be detected. 

None of the control samples contained indicator *E. coli*, except for one sample from which indicator *E. coli* was detected after enrichment of the sample (<10 cfu/g).

When modeling the effects of heat treatment in water baths at constant temperature (see Section 2.5), we saw no robust effect from ESBL status on the proportion of bacteria remaining viable after heat treatment. We saw only a weak and not robustly significant trend towards *lower* survival for ESBL *E. coli* at the very lowest (55 °C) treatment, where bacterial survival rates were nevertheless very high. In general, survival seemed indistinguishable between our genotypes of ESBL *E. coli* and the indicator strains used here. We also saw that the survival time at the higher temperature treatments was very short, and that models using exposure time, temperature and ESBL status explain approximately 90% of the variance, leaving little noise. See Figure 5.

## 4. Discussion

It is mostly unknown how ESBL-producing *E. coli* and other AMR bacteria behave in raw and lightly cooked conditions. Conditions of stress, such as heat, may trigger several mechanisms in bacterial cells, e.g., adaptation, cellular repair, application of response mechanisms and enhanced virulence [19]. Several studies [20,21] have shown that sub-lethal food preservation stresses, such as heat and salt, can significantly alter phenotypic AMR in food-related pathogens such as *E. coli*, *Salmonella Typhimurium*, *Staphylococcus aureus* and *Cronobacter sakazakii*.

For this study we successfully inoculated live blue mussels with *E. coli* by allowing the blue mussels to acclimatize in artificial seawater in the aquarium prior to adding *E. coli.* This is, to our knowledge, the first time an experimental study has been carried out to estimate the survival of *E. coli* as well as ESBLs during the steaming of blue mussels mimicking the cooking procedure in the consumer’s home.

A notable finding is that the study showed that about 1.5% of the *E. coli* present before steaming is expected to survive if the mussels are only steamed until they are opened and not longer, and that the variation between mussels gives an overall likelihood of about 10% to ingest meals with an exposure corresponding to 10% of the pre-cooking bacterial concentration. It is more if half-opened mussels are included.

As the ESBL *E. coli* survived as well as the indicator strains in our other experiments, this means that the traditional preparation method cannot be trusted to inactivate AMR *E. coli* or other bacteria with similar heat inactivation profiles if they are present in the raw mussels. Considering the mussels ability to concentrate bacteria from the surrounding seawater, and that the time point when >95% are open probably represents an optimistic estimate for how long a consumer would keep heat treating, this suggests that they can be a significant source of human exposure to environmental AMR genotypes present in coastal waters or mussel farms, unless effective monitoring and/or heat treatment practices are in place.

The inactivation curves found in our water bath experiments seem consistent with previous reports on *E. coli* heat tolerance [22], where the “shoulder” before inactivation starts is too small to be measured for higher temperatures and probably reflects a combination of heat tolerance and a delay of heat penetrating into the mussel flesh for lower temperatures. No particularly robust “tails” show up in our data, except possibly for the 65 °C treatment where a weak tail effect may be present. For the 55 °C treatment, the inactivation never entered a tail phase, and for 75 °C and steam treatments the inactivation could not be distinguished from log-linear as time progressed. The steaming treatment on the other hand shows a significant “shoulder”, or delay between putting the mussels in the pot and inactivation starting. As shown by the temperature loggers, this likely is due to the time it takes for the inside of mussels to attain a critical temperature seeming to be between 55 °C and 65 °C.

The survival of indicator and ESBL *E. coli* followed indistinguishable trajectories under heat treatment, indicating that thermal inactivation curves for *E. coli* can be used for risk exposure models of resistant isolates, as have been conducted in a recent risk exposure study of ground beef [23], which used the thermal inactivation curve in *E. coli* O157:H7 for hamburgers [23]. Nevertheless, the use of available data needs to be carefully assessed as the different food matrixes and food preparations will have impact of the survival of the specific agent under study [24].

A factor we did not have the opportunity to explore is the differences in heat tolerance between different genotypes of *E. coli*. While *E. coli* is often seen as a heat-sensitive organism, some strains are among the most heat-resistant of foodborne pathogens with D_60_ values >6 min [22,25,26], which suggests inactivation curves with considerably less steep slopes than we observe here are possible. Thus, our model should be treated as a guideline, keeping in mind that judging from the inactivation curves reflecting D_55_ and D_65_ in our experiment, we note that our strain seems to be representative of the most commonly tested of *E. coli* strains [22], but not the most heat tolerant. Further work should take this into account and base risk models for recommendations on inactivation on a wider range of strains found in the relevant environments.

Another factor of concern that needs addressing in this context is the possible transfer of MGEs remaining after thermal inactivation, which cannot be ruled out and needs further studies. Work on post-inactivation MGE transfer and inoculation studies addressing differences in heat tolerance between strains and any possible links between heat resistance and AMR phenotype require further study.

## 5. Conclusions

The present study has indicated that

Shellfish prepared traditionally is a potential pathway for exposure to viable AMR bacteria concentrated from the environment.Inoculation studies mimicking natural bacterial accumulation and realistic preparation have been shown to be feasible and a useful model system.Consuming blue mussels only steamed to opening carries a significant risk of viable bacteria being present in concentrations just one order of magnitude reduced from the raw state.Steaming time (>3.5–4.0 min) is a better indicator than shell openness to avoid exposure to these ESBL or indicator *E. coli* strains.Further studies including more genotypes and relating them to what is found in the environment are needed.No difference in heat tolerance was found between ESBL *E. coli* and an indicator *E. coli* strain in the studied food matrix.

## Figures and Tables

**Figure 1 foods-12-00014-f001:**
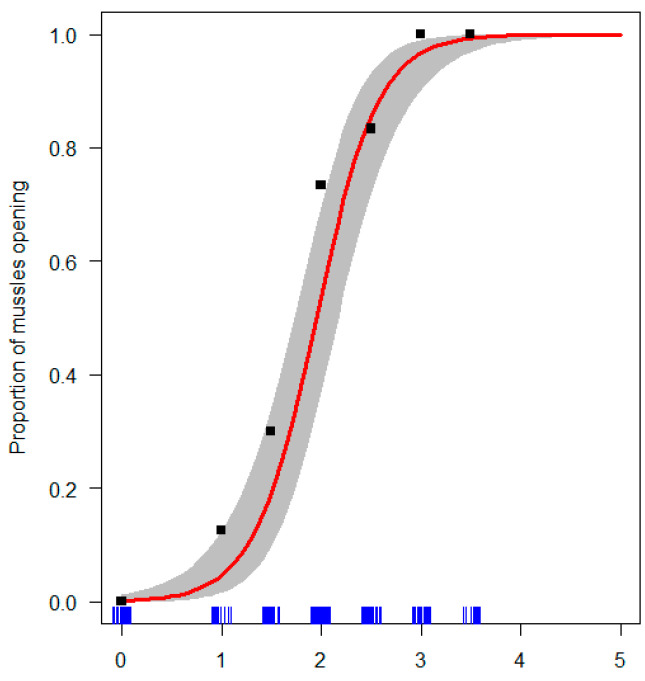
The effect of steaming time on the proportion of mussels opening.

**Figure 2 foods-12-00014-f002:**
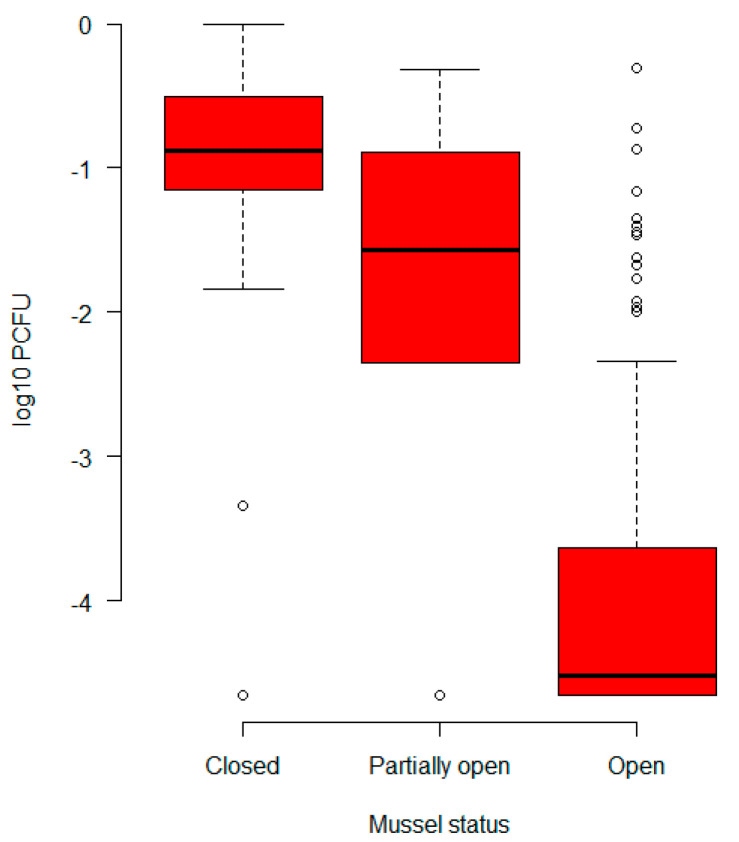
The effect of steam-exposed shell openness status on bacterial concentrations (cfu/g) relative to raw (unopened) mussels (see Equation (2)).

**Figure 3 foods-12-00014-f003:**
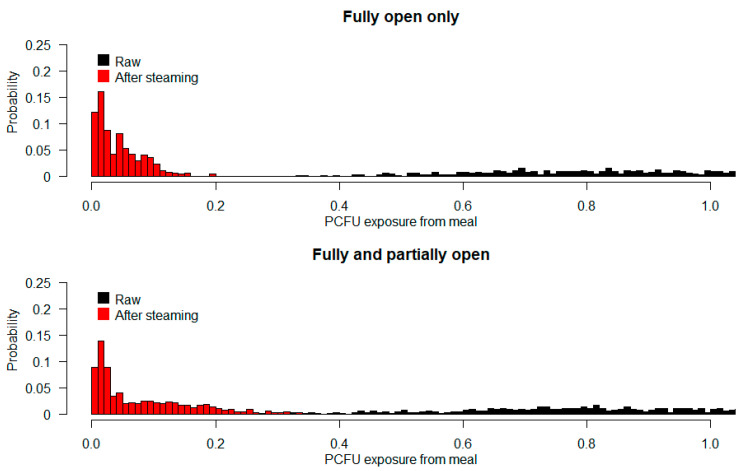
Simulated doses of *E. coli* in a meal size consisting of 20 randomly drawn open mussels (**top** panel) as a proportion of original (un-steamed) contamination (Equation (2)). The red bars are a histogram of such meals consisting of mussels steamed to opening, compared to the load from a similar number of raw shells (black bars). The risk is markedly higher if partially opened mussels are included in the meal at the same frequency they were found in this experiment (**bottom** panel).

**Figure 4 foods-12-00014-f004:**
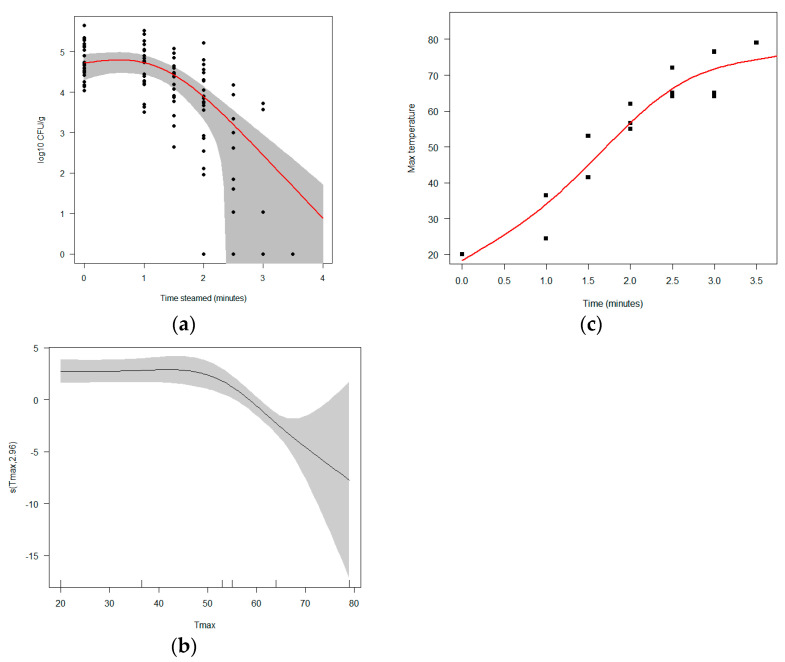
(**a**) Log_10_ bacterial concentrations of *E. coli* as a function of steaming times. The red line gives the best model for inactivation by steaming, with the 2SE confidence interval in gray. (**b**) Maximum temperature registered on loggers glued inside mussel shells as a function of steaming time. (**c**) The effect of maximum temperature on viable bacterial concentration (see equations in Section 2.5). We see that it suggests inactivation from a threshold value a little over 55 °C.

**Figure 5 foods-12-00014-f005:**
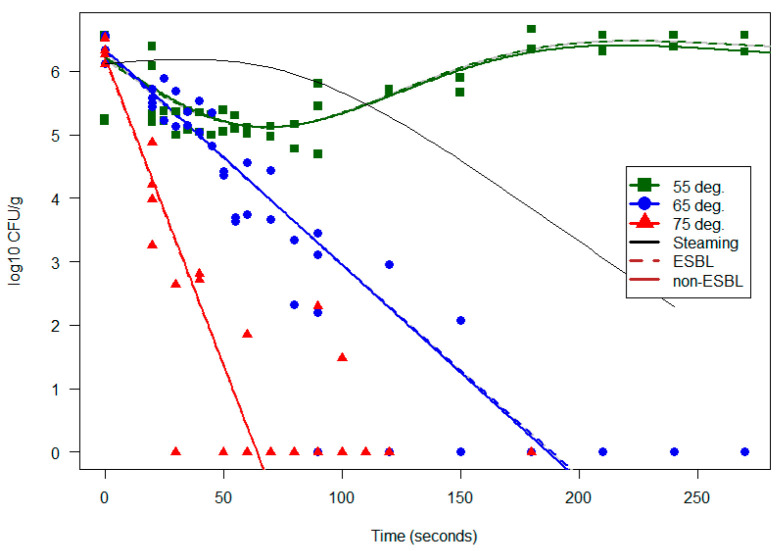
Concentrations (log(10) cfu/g) of *E. coli* and ESBL-producing *E. coli* during heat treatment in inoculated mussel flesh in constant-temperature water baths at 55 °C (green), 65 °C (blue) and 75 °C (red). ESBL and non-ESBL-producing *E.coli* inoculates are shown as solid and broken lines respectively. Only one line indicates overlap.

## Data Availability

Models and research data are available upon request to kyrre.kausrud@vetinst.no.

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
