# Peer review of "The Heat Is On: Modeling the Persistence of ESBL-Producing E. coli in Blue Mussels under Meal Preparation"

_foods, 2022, doi:10.3390/foods12010014_

Round 1

Reviewer 1 Report

The authors evaluate E. coli inactivation on mussels during steaming, comparing generic E. coli and ESBL E. coli. They mimic conditions used by home cooks, and find that steaming until shells are open is not sufficient to inactivate E. coli. Steaming for a designated time is more appropriate. My main concern is that more details are needed in the methods section for someone to effectively reproduce the study.

1. line 116 - provide more information about the strains used, including the strain IDs. 

2. line 119 - which strain? again, strain IDs are important to distinguish those used in the study.

3. line 148 - possibly should be 10^8 CFU ?

4. line 155 - how many times was the experiment repeated?

5. line 157 - what were the time intervals that were used? How many were used?

6. bacterial names should be italicized throughout, please review

7. line 272 - ESBL

Author Response

Thanks for comments. Each of the them has been addressed, as explained below.

  1. line 116 - provide more information about the strains used, including the strain IDs. 

IDs of the strains have been included.

  1. line 119 - which strain? again, strain IDs are important to distinguish those used in the study.

ID has been included

  1. line 148 - possibly should be 10^8 CFU ?

Yes, the text has been changed accordingly.

  1. line 155 - how many times was the experiment repeated?

The experiment was carried out three times. Information has been added.

  1. line 157 - what were the time intervals that were used? How many were used?

The intervals were 30s. Information is added.

  1. bacterial names should be italicized throughout, please review

Done

7. line 272 – ESBL

Reviewer 2 Report

The manuscript is well organized and present important data for food safety (in terms of E. coli and ESBL  producing E. coli in steamed mussels), but contains a number of typos (thus my comment on quality of presentation).

It is suggested to include the information on >3.5 min steaming time in the abstract.

"meat" should be replaced by "flesh", e.g. mussel flesh

lines 15,16: the brackets around ESBL E. coli can be removed.

Please check if bacteria and mussel names are throughout in italics (e.g. lines 14, 34,35,39)

line 40: full stop after [2]

line 42: bacteria have..., line 43: monitoring

line 69: the 230 E. coli are  MPN, this should be mentioned, and the reference is not Reg EC No 854/2004 (which is no longer in force, by the way), but Reg EC No 2073/2005. 

lines 79, 454: mussels

line 101 and others: please check if the word is "acclimatization"

lines 117-118: it would be good the have some strain identification number added.

Materials and methods in general: suppliers of reagents could be given; logger type/brand should  be given, since it must have been a rather small one. Were the loggers calibrated?

line 148: superscript "8"; pr = per?

lines 221-222: maybe in more simple terms: when  E. coli were not detected in quantitive analyses (i.e. <10 cfu/g), but in enrichment culture, E. coli numbers were set to 9 cfu/g.

line 228: Please explain: Pe

line 281: eq. 1-5: does not match with the numbering scheme for equations.

legend to Fig.1: suggest: Black squares are averages. Y-axis values represent fraction of mussels open (i.e. 0 = all closed; 1 = all open)

Please explain SE (standard error) at first mention

figures 2,3: PCFU or PCFU exposure... means what?

line 251: cfug0 means?

line 356: when >3.5 min is "safe", why in the conclusions the statement ">3,5 - 4 min"?

Figs. 3,4: E. coli in italics

line 380: what means "if anything"?

line 421: ESBL-producing E. coli

line 457: "let alone" means?

lines 492ff: the "funding" section is incomplete/not finished.

Author Response

Thanks for comments. Each of the them has been addressed, as explained below.

- It is suggested to include the information on >3.5 min steaming time in the abstract.

The text has been inserted in the abstract

- "meat" should be replaced by "flesh", e.g. mussel flesh

Done

- lines 15,16: the brackets around ESBL E. coli can be removed.

Done

- Please check if bacteria and mussel names are throughout in italics (e.g. lines 14, 34,35,39)

Done

- line 40: full stop after [2]

Done

- line 42: bacteria have..., line 43: monitoring

Changed as requested.

- line 69: the 230 E. coli are  MPN, this should be mentioned, and the reference is not Reg EC No 854/2004 (which is no longer in force, by the way), but Reg EC No 2073/2005.

Thanks for the comment. The criterium for E. coli in shellfish in 2073/2005 is given for shellfish by the end of the manufacturing process, and indicates 1-10 cfu/g. In our text, we referred to the practice to move shellfish to cleaner water before harvest to reach maximum 230 cfu/g. A direct replacement of reference can therefore not be made. However, the paragraph gave been modified to clarify:  

… Commercial produced blue mussels have a food safety regulation limit of maximum 10 E. coli/g by the end of manufacturing process for direct human consumption (Commission regulation (EC) No 2073/2005). Class A shellfish by harvest should not contain more than 230 cfu/g of E. coli by harvest. A recommended practice has been to move shellfish with higher E. coli concentrations to cleaner water until the concentration falls below this level or they will undergo an industrial heat treatment before being marketed [13]. 

- lines 79, 454: mussels

Changed as requested.

- line 101 and others: please check if the word is "acclimatization"

Right, changed as suggested.

- lines 117-118: it would be good the have some strain identification number added.

IDs have been added

- Materials and methods in general: suppliers of reagents could be given; logger type/brand should  be given, since it must have been a rather small one. Were the loggers calibrated?

Producers names have been added. The loggers were calibrated by the producer.

- line 148: superscript "8"; pr = per?

Right. Changed as suggested.

- lines 221-222: maybe in more simple terms: when  E. coli were not detected in quantitive analyses (i.e. <10 cfu/g), but in enrichment culture, E. coli numbers were set to 9 cfu/g.

Thanks for the suggestion. The text is changed accordingly.

- line 228: Please explain: Pe

It means “estimate proportion”. The notation has been changed to be clearer in the text

- line 281: eq. 1-5: does not match with the numbering scheme for equations.

The reference to equations has been adapted. The justification is:

The sentence states “The opening rate is well described by a logistic model of opening as a function of steaming time T which explains about 70% of the variance in openness status for mussels (Methods section 2.2 and eq.1-5, Figure 1).”

Now, eq.  1 describes the opening rate as the estimated proportion (EP) of mussels opening as a function of steaming time (except the equation spelled Time out in full, not just T, -that has now been corrected)

Eq.2 is irrelevant to this sentence, so it has been corrected to “Methods section 2.2 and eq.1,3-5, Figure 1”.

Eq.3,4,5 describe the components of a quasibinomial model used for probability of shell opening.

- legend to Fig.1: suggest: Black squares are averages. Y-axis values represent fraction of mussels open (i.e. 0 = all closed; 1 = all open)

Good suggestion. The figure legend has been changed accordingly

- Please explain SE (standard error) at first mention

Done

- figures 2,3: PCFU or PCFU exposure... means what?

This is defined in eq.2, and the figure legends have been changed to make that clear

- line 251: cfug0 means?

It means mean initial bacterial concentration in the un-steamed mussels, text now rewritten

- line 356: when >3.5 min is "safe", why in the conclusions the statement ">3,5 - 4 min"?

The text in chapter 3.1.2 (line numbers have been shifted, the original line 356 was here)  states that no viable bacteria were recovered from samples steamed >3.5 minutes, but due to the lower detection limit of 10cfu/g  we take the model SE into account when making a recommendation to be sure of erring would be on the side of caution.

- Figs. 3,4: E. coli in italics

Done

- line 380: what means "if anything"?

That if the germination rate had changed at all it had increased, not decreased. The sentence has been rewritten for clarity.

- line 421: ESBL-producing E. coli

Done

- line 457: "let alone" means?

It meant to draw attention to the fact that other AMR bacteria might differ in heat tolerance even more than different strains of ESBL-producing E.coli. However, as the point is not more explored and might be confusing this part of the sentence is superfluous and has been deleted.

- lines 492ff: the "funding" section is incomplete/not finished.

This part is completed